# Meiotic gene silencing complex MTREC/NURS recruits the nuclear exosome to YTH-RNA-binding protein Mmi1

Yuichi Shichino[1☯¤], Yoko Otsubo[1,2,3☯], Masayuki Yamamoto[1,4], Akira Yamashita[1,3,4]*

**1** Laboratory of Cell Responses, National Institute for Basic Biology, Myodaiji, Okazaki, Aichi, Japan,
**2** National Institute for Fusion Science, Toki, Gifu, Japan, **3** Center for Novel Science Initiatives,
National Institutes of Natural Sciences, Myodaiji, Okazaki, Aichi, Japan, **4** Department of Basic Biology,
School of Life Science, SOKENDAI (The Graduate University for Advanced Studies), Myodaiji, Okazaki,
Aichi, Japan

☯ These authors contributed equally to this work.
¤ Current address: RNA Systems Biochemistry Laboratory, Cluster for Pioneering Research, RIKEN,
2–1 Hirosawa, Wako, Saitama, Japan.
* ymst@nibb.ac.jp

journal.pgen.1008598

UNITED KINGDOM

**Data Availability Statement:** All relevant data are
within the manuscript and its Supporting
Information files.

## Abstract

Accurate target recognition in transcript degradation is crucial for regulation of gene expression. In the fission yeast *Schizosaccharomyces pombe*, a number of meiotic transcripts are recognized by a YTH-family RNA-binding protein, Mmi1, and selectively degraded by the nuclear exosome during mitotic growth. Mmi1 forms nuclear foci in mitotically growing cells, and the nuclear exosome colocalizes to such foci. However, it remains elusive how Mmi1 and the nuclear exosome are connected. Here, we show that a complex called MTREC (Mtl1-Red1 core) or NURS (nuclear RNA silencing) that consists of a zinc-finger protein, Red1, and an RNA helicase, Mtl1, is required for the recruitment of the nuclear exosome to Mmi1 foci. Physical interaction between Mmi1 and the nuclear exosome depends on Red1. Furthermore, a chimeric protein involving Mmi1 and Rrp6, which is a nuclear-specific component of the exosome, suppresses the ectopic expression phenotype of meiotic transcripts in *red1Δ* cells and *mtl1* mutant cells. These data indicate that the primary function of MTREC/NURS in meiotic transcript elimination is to link Mmi1 to the nuclear exosome physically.

## Author summary

Since abnormal gene expression is detrimental to proliferation, cells possess a number of mechanisms to regulate gene expression at transcriptional and post-transcriptional levels. In particular, expression of meiotic genes is rigorously repressed in somatic cells because their aberrant expression causes severe cellular defects including genome instability and tumorigenesis. In the fission yeast *Schizosaccharomyces pombe*, selective degradation of meiotic transcripts is employed to prevent their deleterious expression during mitotic

**Funding:** This work was supported by JSPS (https://www.jsps.go.jp/english/index.html) KAKENHI Grant Number 15H04333 to AY, 19K06649 to YO and a grant from the Naito Foundation (https://www.naito-f.or.jp/en/) to YO. YS is a recipient of a JSPS Research Fellowship (PD) 19J00920. The funders had no role in study design, data collection and analysis, decision to publish, or preparation of the manuscript.

**Competing interests:** The authors have declared that no competing interests exist.

growth. Meiotic transcripts are recognized by a YTH-family RNA-binding protein, Mmi1. Mmi1 then induces their selective degradation by the nuclear exosome, which is a highly conserved exonuclease complex. However, little is known how Mmi1 cooperates with the nuclear exosome. Here, we demonstrate that the interaction of Mmi1 with the nuclear exosome is mediated by a complex termed MTREC/NURS that is composed of a conserved zinc-finger protein, Red1, and an RNA helicase, Mtl1. Our findings shed light on the target recognition mechanisms of post-transcriptional regulation.

## Introduction

Stability control of transcripts is crucial to establish appropriate gene expression profiles in a wide range of biological processes [1, 2]. In the fission yeast *Schizosaccharomyces pombe*, expression of a number of meiotic genes is strictly repressed during the mitotic cell cycle by selective RNA elimination, in addition to transcriptional suppression, since their aberrant expression is highly deleterious to cell growth [3, 4]. Selectivity in meiotic RNA degradation is guaranteed by a particular sequence element termed DSR (determinant of selective removal) on meiotic transcripts, which is composed of the repetitive hexanucleotide motif UNAAAC [3, 5–7]. Meiotic transcripts are recognized by a YTH family RNA-binding protein, Mmi1, through direct binding of the YTH domain with UNAAAC DSR motifs [3, 6]. The mode of Mmi1 interaction with RNAs is different from that of other YTH proteins, and Mmi1 does not recognize $N^6$-methyladenosine-containing RNAs [8–11].

Mmi1 induces selective degradation of DSR-containing meiotic transcripts by the nuclear exosome [3, 12]. The exosome is a highly conserved 3′-5′ exonuclease complex that localizes to both the nucleus and cytoplasm [13–15]. The cytoplasmic exosome consists of a nine-subunit inert complex and a catalytically active subunit, Dis3. In the nucleus, an additional exonuclease, Rrp6, binds to the nine-subunit complex in addition to Dis3. Recent genome-wide analyses have revealed that the exosome targets various classes of coding and non-coding transcripts [16–18]. The exosome cooperates with a variety of adapter complexes for selective degradation of divergent target transcripts. Mmi1-mediated selective elimination of meiotic transcripts requires polyadenylation-related factors, including a canonical poly(A) polymerase, Pla1, and a poly(A)-binding protein, Pab2 [12, 19, 20]. Another pivotal factor engaged in Mmi1-mediated RNA degradation is a zinc-finger protein, Red1 [21, 22]. Red1, together with Mtr4-like RNA helicase protein, Mtl1, constitutes a complex termed MTREC (Mtl1-Red1 core) or NURS (nuclear RNA silencing), and mediates the selective degradation of various transcripts, including not only meiotic RNAs, but also cryptic unstable transcripts (CUTs), the latter of which are independent of Mmi1 [23–25]. The machinery of Mmi1-mediated RNA degradation also triggers facultative heterochromatin formation at a subset of its target genes [5, 26, 27]. Furthermore, it regulates transcription termination of its targets [28–30], suggesting that Mmi1-mediated multilayered regulation rigorously shapes appropriate gene expression profiles in mitotically growing cells.

Mmi1 forms nuclear foci in vegetatively growing cells [3]. Most, if not all, factors cooperating with Mmi1, including the nuclear exosome, colocalize with Mmi1 at nuclear foci [12, 21, 22, 24, 31, 32]. We have recently shown that DSR-containing meiotic transcripts are tethered to the Mmi1 foci [33]. Tethering of meiotic transcripts leads to prevention of their nuclear export and deleterious expression of meiotic proteins, even when RNA elimination is compromised [33]. Foci formation of Mmi1 requires its self-interaction with the assistance of a homolog of Enhancer of Rudimentary, Erh1, which induces Mmi1 dimerization [33, 34]. Mmi1

mutant cells lacking the self-interaction domain (SID) and *erh1* deletion mutant cells do not form Mmi1 foci. In these mutant cells, DSR-containing transcripts are liberated from the nuclear foci and escape degradation. These findings highlight the importance of spatial control in Mmi1-mediated regulation. However, it remains largely unknown how factors involved in Mmi1-mediated regulation are incorporated into the nuclear foci. The Red1-containing complex MTREC/NURS binds to both Mmi1 and the nuclear exosome [23–25], suggesting that Red1 acts as a bridge between the nuclear exosome and Mmi1. However, Red1 function in Mmi1-mediated transcript elimination remains to be elucidated.

Here, we show that recruitment of the nuclear exosome to nuclear foci is Red1-dependent. We further demonstrated that Red1 mediates physical interactions between Mmi1 and the nuclear exosome. Our current study sheds new light on our understanding of accurate target recognition in nuclear RNA degradation.

## Results

### Red1 is crucial for nuclear exosome foci formation

To elucidate the mechanisms underlying exosome foci formation, we searched for factors required for the proper localization of Rrp6. We first examined the effect of deletion of the *red1*, *mmi1*, and *pab2* genes, whose gene products colocalize with nuclear foci of Rrp6 [12, 21]. Deletion of *mmi1* causes severe growth defects due to the ectopic expression of meiotic transcripts. Growth defects of *mmi1* deletion can be alleviated by deletion of *mei4*, a crucial target of Mmi1 [3]. Thus, we used a *mmi1Δ mei4Δ* double mutant for examining the impact of *mmi1* deletion. Wild-type cells showed nucleolar localization of Rrp6. Rrp6 also formed foci in both nucleoplasm and nucleolus (Fig 1A and S1A Fig). Strikingly, nuclear foci formation of Rrp6 was severely impaired in *red1Δ* cells, whereas nucleolar accumulation was maintained. Cells lacking *mmi1* exhibited only a modest reduction in the frequency of Rrp6 foci formation and the number of foci per cell, indicating weak contribution of Mmi1 in Rrp6 foci formation. Foci formation of Mmi1 and Red1 was not interdependent (Fig 1A and S1A Fig) [22, 33], and they might play divergent roles in Rrp6 foci formation. The localization of Rrp6 in *pab2Δ* cells was comparable to that observed in wild-type cells. We also found that deletion of *red1* strongly impeded foci formation of both Dis3 and Rrp4, the core components of the exosome (Fig 1B and S1B Fig). Since the fluorescence signal of Dis3 and Rrp4 in the nucleolus in wild-type cells was more intense than that of Rrp6, the frequency of foci formation in or on the border of the nucleolus, and in nucleoplasm, might be underestimated. Deletion of *red1* had no severe impact on nuclear foci formation of Pla1 and Pab2, although percentages of cells containing 1 focus were increased (Fig 1C and 1D, S1C Fig and S1D Fig). From these observations, we concluded that Red1 is a key player in nuclear foci formation of the exosome and regulates the localization of a subset of factors involved in Mmi1-mediated RNA degradation.

### Rrp6 foci formation is dependent on an N-terminal domain of Red1

We next determined which region of Red1 was essential for nuclear foci formation. Since it has been reported that the C-terminal zinc-finger motif is required for the elimination of meiotic transcripts but not for localization [21], we constructed N-terminal deletion mutants according to secondary structure predictions by JPred (Fig 2A) [35]. Comparable expression of the Red1 deletion series was confirmed by western blot analysis (S2A Fig). The deletion of residues 2 to 195 did not alter the localization of Red1 (Fig 2A and 2B). When residues 2 to 347 was deleted, Red1 failed to form nuclear foci and was, rather, uniformly localized in the nucleus. The deletion of residues 196 to 347 also resulted in uniform nuclear localization. We

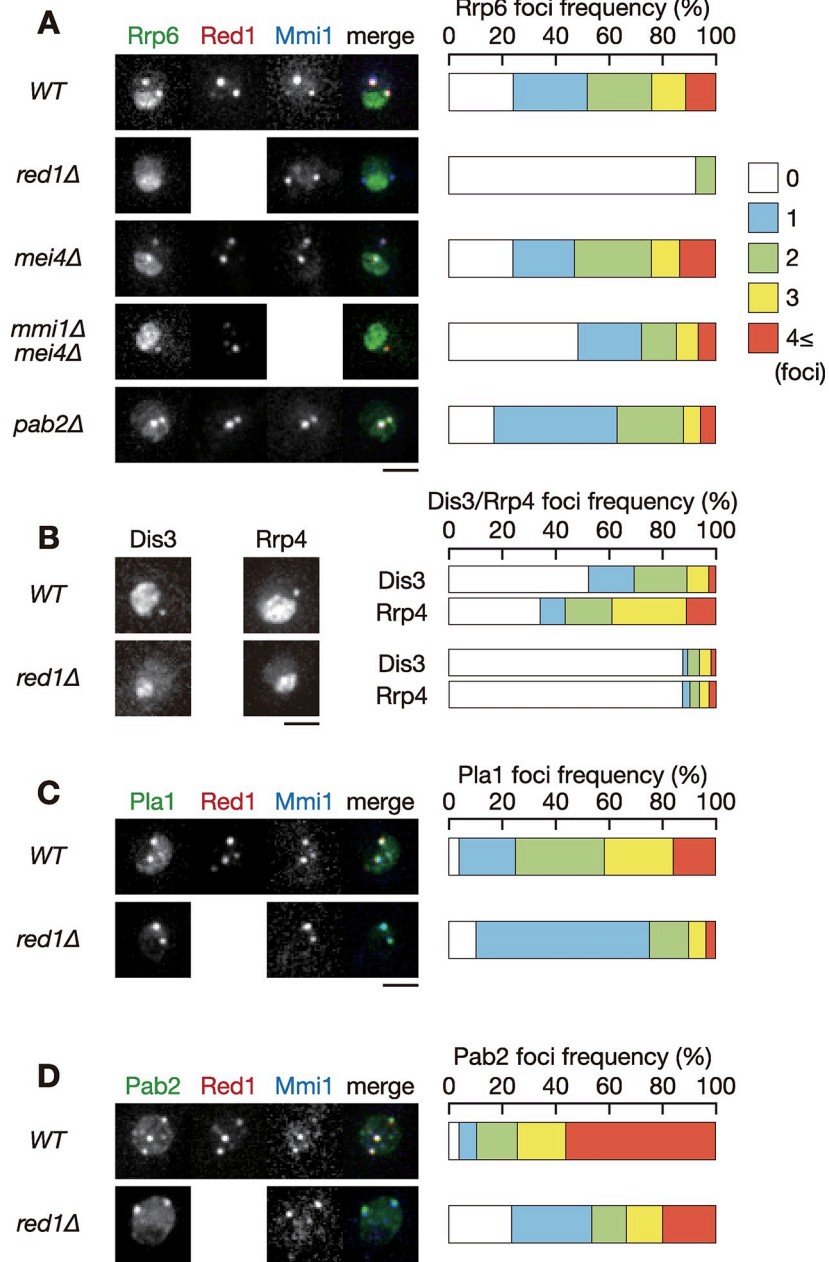

**Fig 1. Red1 is required for nuclear foci formation of exosome.** (**A**) Localization of Rrp6, Red1 and Mmi1 in wild-type, *red1Δ*, *mei4Δ*, *mmi1Δ mei4Δ*, and *pab2Δ* cells. Cells expressing Rrp6-YFP (green), Red1-mCherry (red) or CFP-Mmi1 (blue) from the respective endogenous loci were observed during exponential growth in YE liquid medium. Images of the nuclear region are shown. Frequencies of cells containing 0, 1, 2, 3, or 4 and more Rrp6 foci are indicated on the right (*n* > 100). (**B**) Localization of Dis3 and Rrp4 in wild-type and *red1Δ* cells. *red1Δ* cells expressing Dis3-GFP or Rrp4-GFP from the respective endogenous loci were observed. Frequencies of cells containing 0, 1, 2, 3, or 4 and more Dis3 or Rrp4 foci are indicated (*n* > 100). (**C**) Localization of Pla1, Red1 and Mmi1 in wild-type and *red1Δ* cells. Cells expressing Pla1-YFP (green), Red1-mCherry (red) and CFP-Mmi1 (blue) were examined. Frequencies of cells containing 0, 1, 2, 3, or 4 and more Pla1 foci are indicated (*n* > 100). (**D**) Localization of Pab2, Red1, and Mmi1 in wild-type and *red1Δ* cells. Cells expressing Pab2-YFP (green), Red1-mCherry (red) and CFP-Mmi1 (blue) were examined. Frequencies of cells containing 0, 1, 2, 3, or 4 and more Pab2 foci are indicated (*n* > 100). Scale bars: 2 μm.

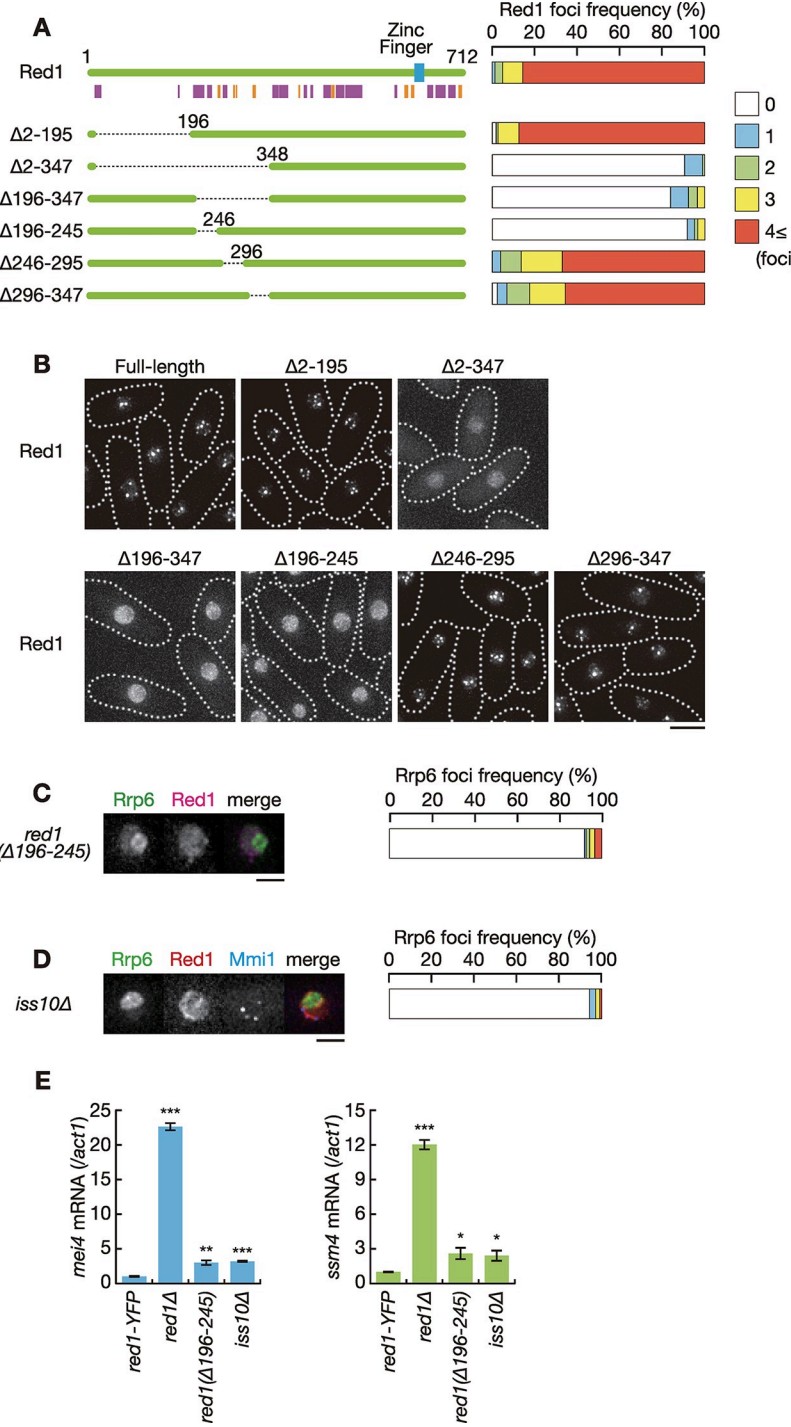

**Fig 2. Red1 forms nuclear foci through the N-terminal domain.** (**A**) Schematic of the structure of Red1 and its truncation series. The blue rectangle represents a putative zinc-finger motif. Predicted alpha helices and beta sheets are shown in magenta and orange, respectively. Frequencies of cells carrying 0, 1, 2, 3, or 4 and more Red1 foci are indicated on the right ($n > 100$). (**B**) Localization of full-length and truncated Red1. YFP-tagged full-length or truncated Red1 were expressed from the endogenous locus. Dotted lines indicate the shape of cells. Scale bar: 5 μm. (**C**) Localization of Rrp6 and Red1 in *red1(Δ196–245)* cells. *red1(Δ196–245)* cells expressing Rrp6-YFP (green) and Red1-mCherry (magenta) from the respective endogenous loci were observed. Images of the nuclear region are shown. Frequency of cells containing 0, 1, 2, 3, or 4 and more Rrp6 foci is indicated on the right ($n > 100$). Scale bar: 2 μm. (**D**) Localization of Rrp6, Red1 and Mmi1 in *iss10Δ* cells. *iss10Δ* cells expressing Rrp6-YFP (green), Red1-mCherry (red) and CFP-Mmi1 (blue) were examined. Frequency of cells containing 0, 1, 2, 3, or 4 and more Rrp6 foci is indicated

($n > 100$). Scale bar: 2 μm. (**E**) Expression of *mei4* mRNA and *ssm4* mRNA in wild-type (*red1-YFP*), *red1Δ*, *red1* (*Δ196–245*), and *iss10Δ* cells. Transcripts were quantified by RT-qPCR and normalized to *act1*. Error bars represent standard error of three independent samples. *$P < 0.05$; **$P < 0.01$; ***$P < 0.001$ compared with the wild-type *red1-YFP* strain (Student's *t*-test).

then divided the domain involving residues 196 to 347 into three regions, and found that residues 196 to 245 were responsible for nuclear foci formation of Red1.

We next examined Rrp6 localization in cells expressing Red1 lacking residues 196 to 245 (*red1(Δ196–245)*), and found that this domain is vital for Rrp6 nuclear foci formation (Fig 2C and S2B Fig). We also tested the effect of deletion of *iss10*, which is essential for Red1 nuclear foci formation [22, 24]. *iss10Δ* cells exhibited a strong reduction in Rrp6 foci formation frequency (Fig 2D and S2C Fig), suggesting that the proper localization of Red1 is crucial for nuclear foci formation of Rrp6.

The growth profile of *red1(Δ196–245)* cells was similar to that of wild-type cells, while *red1Δ* cells grew more slowly than wild-type cells and exhibited severe cold sensitivity (S3A Fig) [21, 22]. *iss10Δ* cells showed mild cold sensitivity, as previously demonstrated [22]. The cold sensitivity of *red1Δ* is not suppressed by deletion of *mei4*, indicating that the defect arises independently of ectopic accumulation of meiotic transcripts [32, 33]. Meiotic transcripts, such as *mei4*, *ssm4* and *rec8* mRNAs, were accumulated in *red1(Δ196–245)* and *iss10Δ* cells during vegetative growth, albeit to a lesser extent than *red1Δ* cells (Fig 2E and S3B Fig). The effect of the deletion of residues 196 to 245 was less prominent on expression of *spo5* mRNAs, which are also targeted by Mmi1. These results suggest that Red1 has some activity without this region or in the absence of Iss10 and that residues 196 to 245 is required to fully induce meiotic transcript degradation.

Red1 has been shown to mediate degradation of CUTs in an Mmi1-independent manner [25]. We then examined the expression of the promoter upstream transcripts (PROMPTs) of the *cti6* gene and the *rpl402* gene, which are known to be CUTs transcribed from the promoter region of *cti6* or *rpl402* in anti-sense direction. In contrast to meiotic transcripts such as *ssm4*, *rec8* and *spo5*, ectopic expression of the *cti6* and *rpl402* PROMPT was detected in *red1Δ* and *rrp6Δ* cells, but not in *mmi1Δ* cells (S3C Fig), as shown previously [25]. *red1(Δ196–245)* and *iss10Δ* cells showed a modest, if any, defect in elimination of the *cti6* and *rpl402* PROMPT, as was the case with meiotic transcripts (S3D Fig). Altogether, the 196–245 domain of Red1 was required for nuclear foci formation of itself and Rrp6, and played a significant role in regulation of meiotic transcript and CUT expression, although it was dispensable for growth at low temperatures.

## Red1 connects Rrp6 with Mmi1 for recruiting target RNAs to the exosome

Red1 physically interacts with both Rrp6 and Mmi1 [21]. To test whether Red1 mediates the interaction between Mmi1 and Rrp6, we carried out co-immunoprecipitation assays. Mmi1 was specifically co-purified with Rrp6 (S4A Fig). RNase treatment did not affect the interaction of Mmi1 with Rrp6 (S4B Fig), indicating that this interaction is independent of RNA molecules. The interaction between these proteins was abolished in the absence of *red1* (Fig 3A), suggesting that Red1 physically links Mmi1 with the exosome. The deletion of residues 196 to 245 in Red1 mildly impaired the interaction between Mmi1 and Rrp6 owing to reduction of the interaction of Red1 with Mmi1 (Fig 3B).

The above results implied that gene expression defects in *red1Δ* cells could be ascribed to impaired interaction between Mmi1 and Rrp6. We investigated this possibility by expressing a chimeric Mmi1-Rrp6 protein in *red1Δ* cells. We constructed a plasmid carrying a chimeric

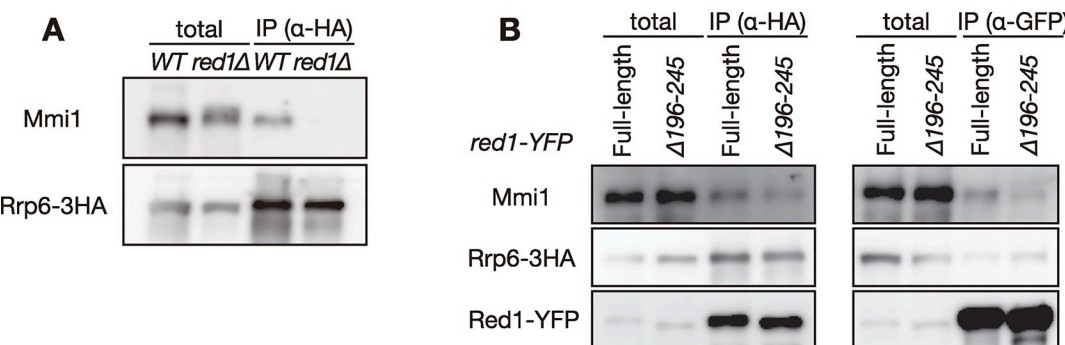

**Fig 3. Red1 is required for interaction between Rrp6 and Mmi1.** (**A**) Co-immunoprecipitation of Rrp6 and Mmi1 in wild-type and *red1Δ* cells. Native cell extracts were prepared from exponentially growing cells expressing Rrp6-3HA in liquid MM medium and subjected to immunoprecipitation with anti-HA antibody. Precipitates and 10% total cell extracts were then immunoblotted with anti-Mmi1 and anti-HA antibodies. (**B**) Co-immunoprecipitation of Rrp6 and Mmi1 (left), or Red1 and Mmi1 (right) in wild-type and *red1(Δ196–295)* cells. Native cell extracts were subjected to immunoprecipitation with anti-HA antibody or anti-GFP antibody. Precipitates and 10% total cell extracts were then immunoblotted with anti-Mmi1, anti-HA and anti-GFP antibodies.

gene in which GFP was inserted between full-length *rrp6* and *mmi1* (*rrp6-GFP-mmi1*) ORFs. The chimeric protein Rrp6-GFP-Mmi1 was able to suppress the growth defect, and ectopic accumulation of meiotic transcripts and CUTs in *rrp6Δ* cells (S5A and S5B Fig). Rrp6-GFP-Mmi1 also suppressed the growth defect and ectopic accumulation of meiotic transcripts in temperature-sensitive *mmi1* mutant cells (*mmi1-ts3*) (S5C and S5D Fig). In these experiments, cells were cultured in minimal medium to induce expression of the chimeric protein. Since *mmi1Δ* cells show growth retardation in minimal medium even when combined with the *mei4* deletion, we used the temperature-sensitive *mmi1* mutant instead of *mmi1Δ*. These data indicated that Rrp6-GFP-Mmi1 maintains the functions of both Rrp6 and Mmi1. Intriguingly, aberrant accumulation of *mei4*, *ssm4*, *rec8* and *spo5* mRNAs in *red1Δ* cells was abrogated when Rrp6-GFP-Mmi1 was expressed, whereas not by either Rrp6 or Mmi1 alone (Fig 4A and S6A Fig). It should be noted that the use of minimal medium in these experiments resulted in the lower accumulation of meiotic transcripts than that in rich medium (S3 Fig). Meanwhile, *cti6* and *rpl402* PROMPTs still accumulated in *red1Δ* cells even when Rrp6-GFP-Mmi1 was expressed (Fig 4A and S6A Fig). Rrp6-GFP-Mmi1 did not suppress the cold sensitivity of *red1Δ* cells (S6B Fig), consistently with that the growth defect caused by *red1* deletion is irrelevant to ectopic expression of meiotic transcripts [32, 33]. We found that the slight growth retardation of *red1Δ* cells at 25 to 36°C, which has been shown previously [21, 22], was suppressed by expression of Rrp6, Mmi1 or Rrp6-GFP-Mmi1 as well as Red1, although the underlying mechanism remains unclear.

We next confirmed the Red1-mediated interaction between Mmi1 and Rrp6 by *in vitro* binding assay (Fig 4B). The weak direct interaction of Mmi1 with Rrp6 was observed in our experimental conditions. Addition of Red1 greatly increased the interaction, indicating the role of Red1 in connecting Mmi1 to the exosome. The direct binding of Red1 with Rrp6 was also observed (Fig 4B).

From these results, we concluded that the physical interaction between Rrp6 and Mmi1 is crucial for the elimination of meiotic transcripts, but not for other function(s) involving Red1.

## Self-interaction of Mmi1 is vital for meiotic transcript elimination by Rrp6

We previously demonstrated that Mmi1 interacts with itself via the self-interaction domain, SID, which is essential for the proper function and localization of Mmi1 [33]. Given that

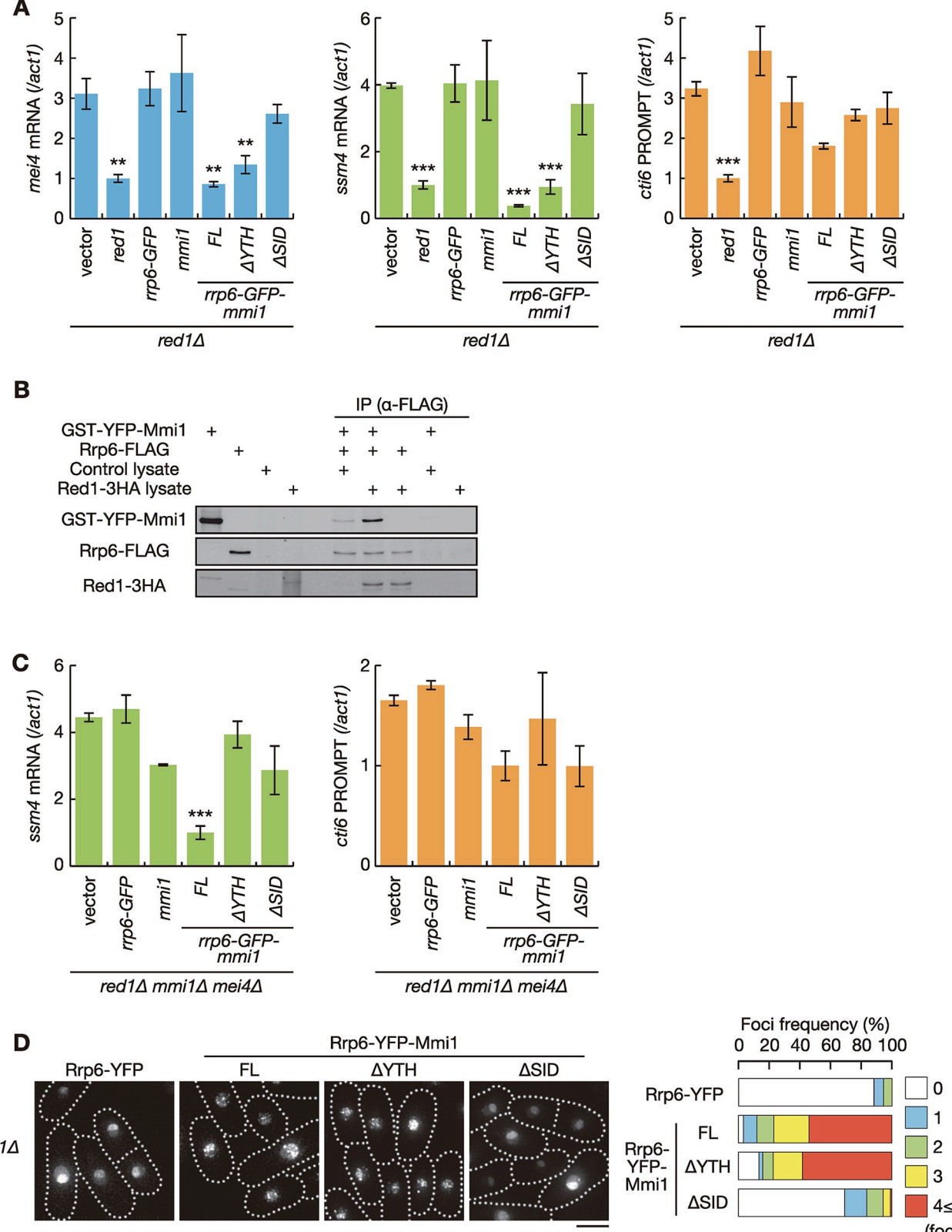

**Fig 4. Interaction between Rrp6 and Mmi1 via Red1 is essential for meiotic transcript elimination.** (A) Expression of *mei4* mRNA, *ssm4* mRNA and *cti6* PROMPT in *red1Δ* cells expressing Red1, Rrp6-GFP, Mmi1 or chimeric proteins composed of Rrp6, GFP, and full-length or truncated Mmi1 from

plasmids in liquid MM medium. Transcripts of each gene were analyzed by RT-qPCR and normalized to *act1*. Error bars represent standard error of three independent samples. \*\**P* < 0.01; \*\*\**P* < 0.001 compared with cells carrying empty vector (Student's *t*-test). (**B**) *In vitro* interaction between Mmi1 and Rrp6. Purified Rrp6-FLAG was incubated with purified GST-YFP-Mmi1 and reticulocyte lysate expressing Red1-3HA or control lysate, and was precipitated with anti-FLAG antibody. Precipitates were then immunoblotted with anti-GFP, anti-HA and anti-FLAG antibodies. (**C**) Expression of *ssm4* mRNA and *cti6* PROMPT in *red1Δ mmi1Δ mei4Δ* cells expressing Red1, Rrp6-GFP, Mmi1, or chimeric proteins composed of Rrp6, GFP, and full-length or truncated Mmi1 from plasmids in liquid MM medium. Transcripts were analyzed by RT-qPCR and normalized to *act1*. Error bars represent standard error of three independent samples. \*\*\**P* < 0.001 compared with cells carrying empty vector (Student's *t*-test). (**D**) Localization of Rrp6-Mmi1 fusion proteins. *red1Δ* cells expressing Rrp6-YFP or chimeric proteins composed of Rrp6, YFP, and full-length or truncated Mmi1 from plasmids were observed during exponential growth. Frequencies of cells containing 0, 1, 2, 3, or 4 and more nuclear foci are indicated on the right (*n* > 100). Dotted lines indicate the shape of cells. Scale bar: 5 μm.

Rrp6-GFP-Mmi1 functions by interacting with endogenously expressed Mmi1, the YTH-RNA-binding domain in the chimeric protein could be unnecessary for meiotic transcript elimination because target transcripts might be recognized by endogenous Mmi1 and delivered to Rrp6 in the chimeric protein through Mmi1-Mmi1 interaction. To test this hypothesis, we constructed a plasmid carrying a fusion gene of *rrp6*, *GFP* or *YFP*, and *mmi1-ΔYTH* or *mmi1-ΔSID*, lacking the region encoding the YTH-RNA-binding domain or the SID, respectively. We confirmed that chimeric proteins were expressed in a comparable amount, although removal of the SID resulted in an increase of the expression level (S6C Fig). Both chimeric proteins maintained the function of Rrp6 (S5A and S5B Fig), but lost the ability to act as Mmi1 (S5C and S5D Fig). The chimeric protein containing Mmi1-ΔYTH showed a negative effect in *mmi1-ts3* cells at permissive temperatures (S5C and S5D Fig). This is because Mmi1 lacking the YTH domain exerts a dominant-negative activity, as shown previously [33]. Rrp6-GFP-Mmi1-ΔYTH amended the degradation defect of *mei4*, *ssm4*, *rec8* and *spo5* mRNAs in *red1Δ* cells, but Rrp6-GFP-Mmi1-ΔSID did not (Fig 4A and S6A Fig). Rrp6-GFP-Mmi1-ΔYTH did not suppress the degradation defect in *red1Δ mmi1Δ* cells (Fig 4C and S6D Fig), indicating that Rrp6-GFP-Mmi1-ΔYTH requires endogenous Mmi1 to exert the function in the absence of Red1. The interaction between Rrp6-GFP-Mmi1 and endogenous Mmi1 was demonstrated by co-immunoprecipitation (S6E Fig). This interaction was dependent on the SID but not on the YTH domain, as predicted.

We next examined the localization of the chimeric proteins. YFP instead of GFP was used for the observation to accommodate the microscope filter set. Rrp6-YFP-Mmi1 formed nuclear foci in *red1Δ* cells (Fig 4D). The absence of the YTH domain did not affect the localization of the chimeric protein. However, SID deletion significantly reduced the frequency of foci formation. These observations suggested that Mmi1 self-interaction is crucial for foci formation of the chimeric proteins, and for meiotic transcript degradation by Rrp6, although we cannot exclude the possibility that the SID is required for the interaction of Mmi1 with other factors required for meiotic RNA elimination.

## Rrp6 is required for foci formation of other exosome components

Rrp6 mediates the interaction between core components of the exosome and various exosome-related factors [36]. To examine whether Rrp6 acts as a mediator between Red1 and other exosome components, we observed the localization of core components of the exosome in *rrp6Δ* cells. Dis3 and Rrp4 failed to form foci in the absence of *rrp6* (Fig 5A and S7A Fig), while Red1 and Mmi1 showed similar localization to wild-type cells (Fig 5B and S7B Fig). In *red1Δ* cells, in which foci formation of Rrp6, Dis3 and Rrp4 was severely impaired (Fig 1A and 1B), Rrp6-YFP-Mmi1 partially recovered the foci formation frequencies of Dis3 and Rrp4 (Fig 5C and S7C Fig). Foci formation frequency was similar even when the YTH domain was absent. The recovery of Dis3 and Rrp4 foci formation depended on the SID of Mmi1 in the

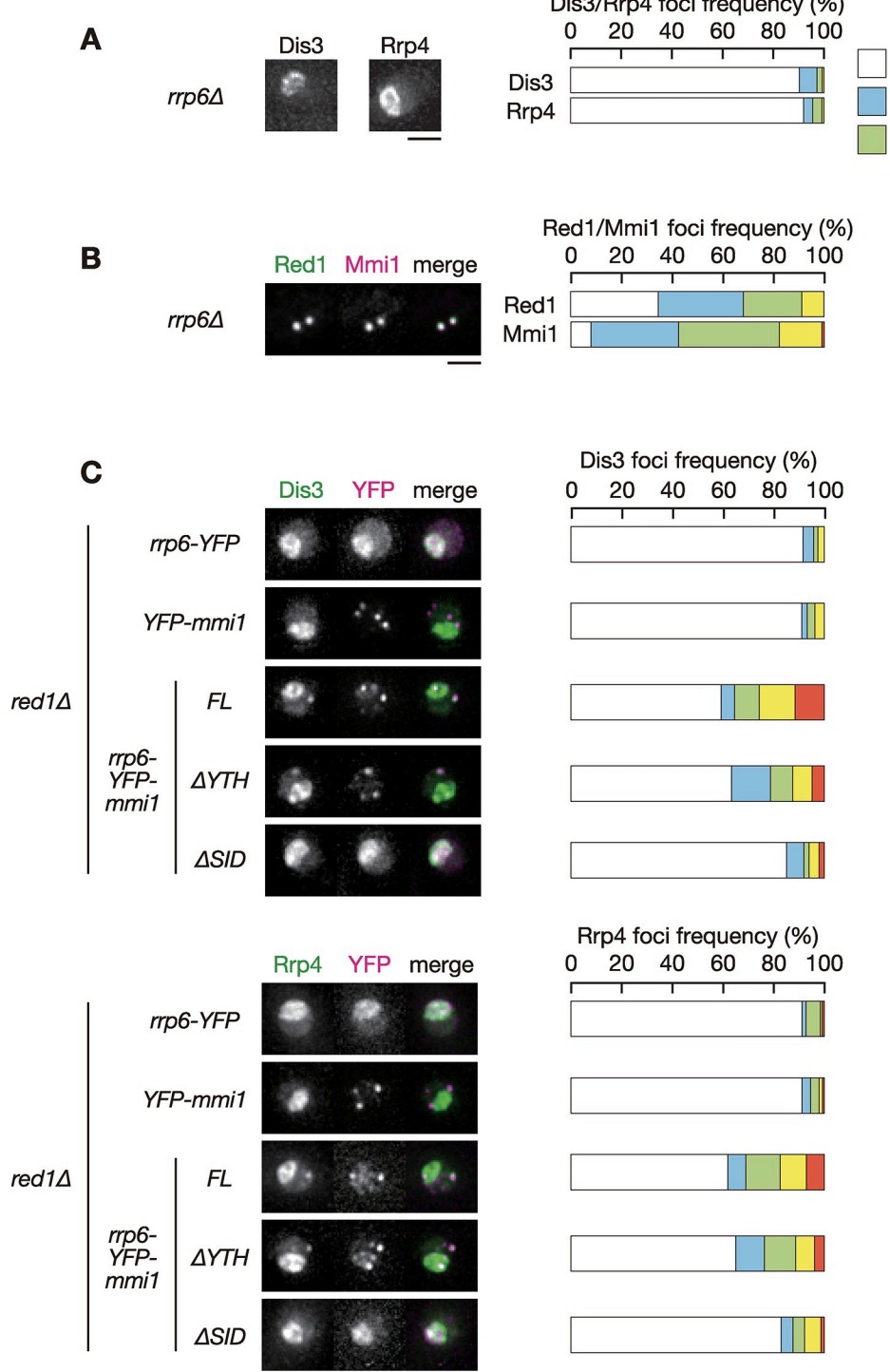

**Fig 5. Rrp6 is vital for nuclear foci formation of other exosome components.** (**A**) Localization of Dis3 and Rrp4 in *rrp6Δ* cells. *rrp6Δ* cells expressing Dis3-GFP or Rrp4-GFP from the respective endogenous loci were observed during exponential growth in liquid YE medium. Images of the nuclear region are shown. Frequencies of cells containing 0, 1, 2, 3, or 4 and more Dis3 or Rrp4 foci are indicated on the right (*n* > 100). (**B**) Localization of Red1 and Mmi1 in *rrp6Δ* cells. *rrp6Δ* cells expressing Red1-YFP (green) and CFP-Mmi1 (magenta) were examined. Frequencies of cells containing 0, 1, 2, 3, or 4 and more Red1 or Mmi1 foci are indicated on the right (*n* > 100). (**C**) Localization of Dis3 and Rrp4 in *red1Δ* cells expressing Rrp6-YFP, YFP-Mmi1, or chimeric proteins composed of Rrp6, YFP, and full-length or truncated Mmi1. Dis3-mCherry and Rrp4-mCherry were expressed from the respective endogenous loci (green) and YFP-containing chimeric proteins

were expressed from plasmids (magenta) in liquid SD medium. Frequencies of cells containing 0, 1, 2, 3, or 4 and more Dis3 or Rrp4 are indicated on the right ($n > 100$). Scale bars: 2 μm.

chimeric protein. These results suggested that Rrp6 plays a key role in the assembly of the exosome at nuclear foci containing Mmi1 and Red1.

### Red1 links Mmi1 to the exosome in cooperation with Mtl1

We next investigated whether the function of Red1 to connect Mmi1 with the exosome required Mtl1, a component of the Red1-containing MTREC/NURS complex [23, 24]. Nuclear foci formation of Rrp6 was compromised in *mtl1* mutant cells (Fig 6A). The relatively mild effect compared to that in *red1Δ* cells (Fig 1A) might arise from the use of a conditional cold-sensitive mutant, since Mtl1 is essential for cell growth. We further examined the expression of meiotic transcripts in the *mtl1-cs5* cells expressing Rrp6-GFP-Mmi1. Expression of the chimeric protein suppressed aberrant accumulation of meiotic mRNAs as efficiently as that of wild-type Mtl1 (Fig 6B and S8 Fig). These observations suggest that Mtl1 contributes to connect Mmi1 and the nuclear exosome by forming the MTREC/NURS complex with Red1.

## Discussion

In this study, we demonstrated that the primary function of Red1 in meiotic transcript elimination is to connect Mmi1 and the nuclear exosome physically. We have also shown that Rrp6 is crucial for nuclear foci formation of the core exosome. This observation is consistent with the result of previous co-purification assays, which showed that the interaction between exosome subunits and Red1 depends on Rrp6 [25]. Red1 forms the complex, MTREC or NURS, which includes an RNA helicase, Mtl1 [23, 24]. Our observations also imply that the MTREC/NURS complex, rather than Red1 alone, may link Mmi1 to the exosome.

We found that foci formation of Rrp6 was severely impaired by the deletion of residues 196 to 245 in Red1. Meanwhile, ectopic expression levels of meiotic transcripts were moderate in *red1(Δ196–245)* cells compared to those in *red1Δ* cells. These results suggest that nuclear foci, to which Rrp6 and Mmi1 localize, might not be the exclusive sites for Mmi1-mediated RNA degradation. Indeed, the localization of factors involved in Mmi1-mediated degradation is not limited to nuclear foci; Mmi1, Red1 and Rrp6 have been shown to localize to the loci encoding target transcripts including *mei4* by chromatin immunoprecipitation experiments [23, 24, 26, 27, 32, 37], although their colocalization with the *mei4* locus was not observed by microscopic observations [24, 37]. Thus, it is plausible that Mmi1-mediated RNA degradation also takes place at the genetic loci where its targets are transcribed. Further clarification of the site(s) of RNA degradation would be important for understanding of Mmi1-mediated regulation.

Red1 exerts various functions beyond meiotic transcript elimination. For instance, Red1 plays an essential role in degradation of CUTs by the nuclear exosome, independently of Mmi1 [25]. Red1 is also required for growth at low temperatures [21, 22], although it remains elusive what Red1 does under cold conditions. It is interesting whether Red1 physically links a target-recognition factor to the degradation machinery, such as the nuclear exosome, in other situations as in meiotic transcript elimination. Another intriguing question is whether Red1 function is conserved in other organisms. A human zinc-finger protein, ZFC3H1, which is suggested to be a counterpart of Red1, forms a complex termed PAXT (poly(A) tail exosome targeting) together with an Mtl1-ortholog, hMtr4, and plays an important role in the selective elimination of polyadenylated nuclear RNAs [38, 39]. Intriguingly, ZFC3H1 is required not only for RNA degradation but also retention of target transcripts to nuclear foci [39, 40]. We

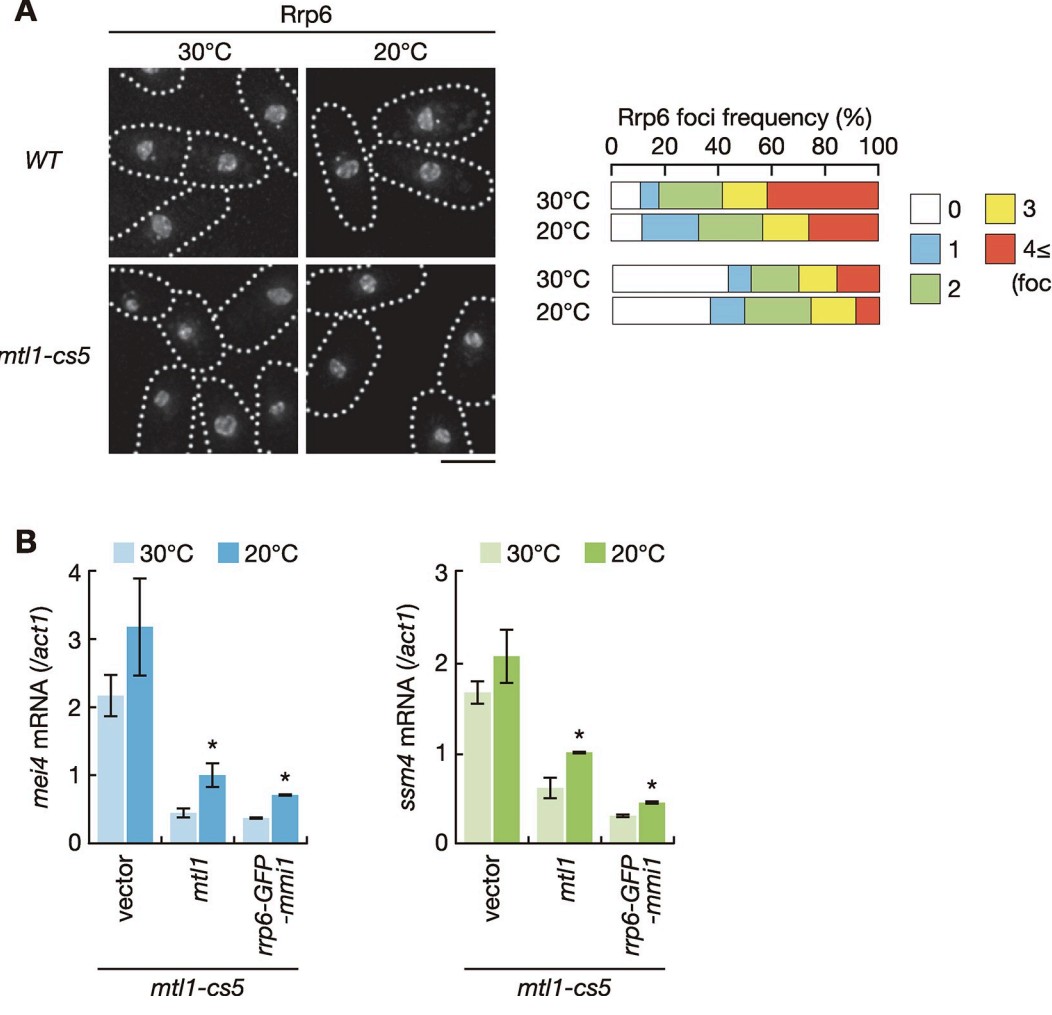

**Fig 6. Mtl1 mediates interaction between Rrp6 and Mmi1 to eliminate meiotic transcripts.** (**A**) Localization of Rrp6 in wild-type and *mtl1-cs5* cells. Cells expressing Rrp6-mCherry from the endogenous locus were observed during exponential growth at 30°C in liquid YE medium and shifted to 20°C for 2 hours. Dotted lines indicate the shape of cells. Frequencies of cells containing 0, 1, 2, 3, or 4 and more Rrp6 foci are indicated on the right ($n > 100$). Scale bar: 5 μm. (**B**) Expression of *mei4* mRNA and *ssm4* mRNA in *mtl1-cs5* cells expressing Mtl1 and Rrp6-GFP-Mmi1 from plasmids. Cells were grown in liquid MM medium at 30°C and shifted to 20°C for 2 hours. Transcripts were quantified by RT-qPCR and normalized to *act1*. Error bars represent standard error of three independent samples. *$P < 0.05$ compared with cells carrying empty vector at 20°C (Student's *t*-test).

have demonstrated that Red1 is dispensable for tethering of meiotic transcripts to nuclear foci, while Mmi1 prevents nuclear export of its targets by sequestering them to foci even when RNA degradation is dampened [33]. Future studies may clarify the conservation of Red1 function in higher eukaryotes.

# Materials and Methods

## Yeast strains and general genetic methods

The *S. pombe* strains used in this study are listed in S1 Table. The general genetic manipulation procedures and growth media have been previously described [41, 42]. Deletion mutants and epitope-tagged strains were constructed using PCR-based gene-targeting protocols [43, 44].

To construct truncated *red1* mutant strains, we cloned the *red1* open reading frame (ORF) with 1 kb upstream and downstream flanking regions and removed the indicated regions using the PrimeStar Mutagenesis kit (Takara, Shiga, Japan). PCR-amplified DNA fragments encompassing each truncated *red1* ORF with flanking regions were introduced into the *red1::ura4⁺* strain. Transformants were counter-selected on medium containing 5-fluoroorotic acid.

Plasmids expressing chimeric *rrp6-GFP-mmi1* gene and its variants were constructed by cloning the PCR-amplified *rrp6* and full-length or truncated *mmi1* ORFs into pREP81 carrying the green fluorescent protein (GFP) ORF, in which the fusion genes were expressed under the control of a modified thiamine-repressive *nmt1* promoter [45]. To observe the localization and expression levels of chimeric proteins (Figs 4D and 5C, S6C Fig and S7C Fig), GFP was replaced with yellow fluorescent protein (YFP) and the fusion genes were expressed from the modified constitutively active *adh1* promoter.

### Fluorescence microscopy

Cells were grown in logarithmic phase at 30°C and mounted onto lectin-coated glass bottom culture dishes (MatTek, Ashland, MA) filled with liquid growth medium. Images were acquired using the DeltaVision-SoftWoRx system (GE Healthcare, Chicago, IL) by collecting 12 optical sections along the z-axis at 0.5 μm intervals. All images were deconvolved and merged into single projections using SoftWoRx software.

### RNA preparation and reverse-transcription quantitative PCR (RT-qPCR) analysis

Total RNA extraction and RT-qPCR analysis were performed as previously described [33] by using ReverTra Ace qPCR Master Mix (TOYOBO, Osaka, Japan) and a LightCycler96 instrument (Roche, Basel, Switzerland) with SYBR Premix Ex Taq II (Takara). Normalization was performed using *act1*, which encodes actin. Primers used in this study are listed in S2 Table.

### Immunoprecipitation and western blot analysis

Immunoprecipitation was performed as previously described [46]. Hemagglutinin (HA)-tagged Rrp6 was precipitated using mouse anti-HA-tag magnetic beads M180-11 (MBL, Nagoya, Japan). Red1-YFP and GFP-containing chimeric proteins were precipitated and detected using anti-GFP-tag magnetic beads D153-11 (MBL) and anti-GFP antibody [47], respectively. Rabbit anti-Mmi1 TB0514 (our laboratory preparation) and mouse anti-HA 12CA5 (Sigma-Aldrich, St. Louis, MO) were used to detect Mmi1 and Rrp6-3HA, respectively. For immunoprecipitation with RNase treatment, cell lysates were incubated for 30 min at room temperature with 0.5 U of RNase A and 20 U of RNase T1 (RNase Cocktail Enzyme Mix, Thermo Fisher Scientific, Waltham, MA). In S2A Fig and S6C Fig, harvested cells were disrupted with glass beads in 20% trichloroacetic acid. Anti-GFP antibody [47] was used to detect Red1-YFP, Rrp6-YFP and Rrp6-YFP-Mmi1. Anti-γ-tubulin GTU-88 (Sigma-Aldrich) was used for a loading control.

### *in vitro* binding assay

PCR-amplified *YFP-mmi1* and *rrp6-FLAG* were cloned into pCold-GST (Takara). Glutathione S-transferase (GST)-tagged YFP-Mmi1 and Rrp6-FLAG proteins purified from *E. coli* were separated by NGC Chromatography System (Bio-rad, Hercules, CA) with HiTrap Heparin HP and HiTrap Q HP column (GE Healthcare) for GST-YFP-Mmi1 and GST-Rrp6-FLAG, respectively. It should be noted that most of GST-Rrp6-FLAG was cleaved between GST and

Rrp6-FLAG during the preparation. HA-tagged Red1 was expressed in Rabbit Reticulocyte Lysate System, Nuclease Treated (Promega, Madison, WI) according to the manufacturer's instruction. mRNA containing the His-tagged *red1-3HA* gene was transcribed using T7-Scribe Standard RNA IVT Kit (Cellscript, Madison, WI). Capping and poly(A)-tailing were performed using ScriptCap m$^7$G Capping System (Cellscript), ScriptCap 2'-O-Methyltransferase Kit (Cellscript), and A-Plus Poly(A) Polymerase Tailing Kit (Cellscript). For negative controls, we parallelly performed *in vitro* translation without mRNA. Proteins were incubated at the room temperature for 30 min in the binding buffer (20 mM HEPES pH 7.5, 150 mM NaCl, 5 mM MgCl$_2$, 1 mM dithiothreitol (DTT), 0.5% Triton-X100, 5% glycerol), and then mixed with Anti-DDDDK-tag mAb-Magnetic Beads M185-11 (MBL) at 4˚C for 30 min. Beads were washed by wash buffer (20 mM HEPES pH 7.5, 300 mM NaCl, 5 mM MgCl$_2$, 1 mM DTT, 1% Triton-X100) for 5 times, and subjected to western blotting. Rabbit anti-GFP ab6556 (abcam, Cambridge, UK), mouse anti-FLAG M2 (Sigma-Aldrich), and rat anti-HA 3F10 (Sigma-Aldrich) were used to detect GST-YFP-Mmi1, Rrp6-FLAG, and Red1-3HA, respectively. Images were acquired by Odyssey CLx Infrared Imaging System (LI-COR, Lincoln, NE).

## Supporting information

**S1 Fig. Red1 is required for nuclear foci formation of exosome.** (**A**) Localization of Rrp6, Red1 and Mmi1 in wild-type, *red1Δ*, *mei4Δ*, *mmi1Δ mei4Δ*, and *pab2Δ* cells. Cells expressing Rrp6-YFP (green), Red1-mCherry (red) or CFP-Mmi1 (blue) from the respective endogenous loci were observed during exponential growth in YE liquid medium. Dotted lines indicate the shape of cells. Boxed regions are magnified in Fig 1A. (**B**) Localization of Dis3 and Rrp4 in wild-type and *red1Δ* cells. *red1Δ* cells expressing Dis3-GFP or Rrp4-GFP from the respective endogenous loci were observed. Boxed regions are magnified in Fig 1B. (**C**) Localization of Pla1, Red1 and Mmi1 in wild-type and *red1Δ* cells. Cells expressing Pla1-YFP (green), Red1--mCherry (red) and CFP-Mmi1 (blue) were examined. Boxed regions are magnified in Fig 1C. (**D**) Localization of Pab2, Red1, and Mmi1 in wild-type and *red1Δ* cells. Cells expressing Pab2-YFP (green), Red1-mCherry (red) and CFP-Mmi1 (blue) were examined. Boxed regions are magnified in Fig 1D. Scale bars: 5 μm.
(PDF)

**S2 Fig. Red1(Δ196–245) is defective in Rrp6 foci formation.** (**A**) Expression levels of truncated Red1 proteins. Cell extracts were prepared from exponentially growing cells expressing wild-type or truncated Red1-YFP in liquid YE medium and immunoblotted with anti-GFP antibody. γ-tubulin was used as a loading control. The asterisks indicate non-specific bands. (**B**) Localization of Rrp6 and Red1 in *red1(Δ196–245)* cells. *red1(Δ196–245)* cells expressing Rrp6-YFP (green) and Red1-mCherry (magenta) from the respective endogenous loci were observed. Dotted lines indicate the shape of cells. Boxed region is magnified in Fig 2C. Scale bar: 5 μm. (**C**) Localization of Rrp6, Red1 and Mmi1 in *iss10Δ* cells. *iss10Δ* cells expressing Rrp6-YFP (green), Red1-mCherry (red) and CFP-Mmi1 (blue) were examined. Boxed region is magnified in Fig 2D. Scale bar: 5 μm.
(PDF)

**S3 Fig. Red1(Δ196–245) is defective in meiotic transcript degradation.** (**A**) Growth profiles of wild-type (*red1-YFP*), *red1Δ*, *iss10Δ* and *red1(Δ196–245)* cells. Ten-fold serial dilutions of cells were spotted on YE medium and incubated at the indicated temperatures. (**B**) Expression of *rec8* mRNA and *spo5* mRNA in wild-type (*red1-YFP*), *red1Δ*, *iss10Δ* and *red1(Δ196–245)* cells. Transcripts were quantified by RT-qPCR and normalized to *act1*. Error bars represent standard error of three independent samples. $^*P < 0.05$; $^{***}P < 0.001$ compared with the wild-

type *red1-YFP* strain (Student's *t*-test). (**C**) Expression of *ssm4* mRNA, *rec8* mRNA, *spo5* mRNA, *cti6* PROMPT and *rpl402* PROMPT in wild-type, *mmi1Δ mei4Δ*, *red1Δ* and *rrp6Δ* cells. Transcripts were quantified by RT-qPCR and normalized to *act1*. Error bars represent standard error of three independent samples. *$P < 0.05$; **$P < 0.01$; ***$P < 0.001$ compared with the wild-type strain (Student's *t*-test). (**D**) Expression of *cti6* PROMPT and *rpl402* PROMPT in wild-type (*red1-YFP*), *red1Δ*, *red1(Δ196–245)*, and *iss10Δ* cells. Transcripts were quantified by RT-qPCR and normalized to *act1*. Error bars represent standard error of three independent samples. *$P < 0.05$; **$P < 0.01$; ***$P < 0.001$ compared with the wild-type *red1--YFP* strain (Student's *t*-test).
(PDF)

**S4 Fig. Rrp6 physically interacts with Mmi1.** (**A**) Co-immunoprecipitation of Rrp6 and Mmi1 in wild-type cells. Native cell extracts were prepared from exponentially growing cells expressing Rrp6-3HA from a plasmid in MM medium and subjected to immunoprecipitation with anti-HA antibody. Cells carrying an empty vector were used as a negative control. Precipitates and 10% total cell extracts were then immunoblotted with anti-Mmi1 and anti-HA antibodies. (**B**) The effect of RNase treatment on interaction between Rrp6 and Mmi1. Native cell extracts were incubated with or without RNase and subjected to immunoprecipitation with anti-HA antibody.
(PDF)

**S5 Fig. Chimeric Rrp6-GFP-Mmi1 protein maintains the function of Rrp6 and Mmi1.** (**A**) Growth profiles of *rrp6Δ* cells expressing Red1, Rrp6-GFP, Mmi1, or chimeric proteins composed of Rrp6, GFP, and full-length or truncated Mmi1 from plasmids. Ten-fold serial dilutions of cells were spotted on MM medium and incubated at the indicated temperatures. (**B**) Expression of *mei4* mRNA, *ssm4* mRNA, and *cti6* PROMPT in *rrp6Δ* cells expressing Red1, Rrp6-GFP, Mmi1, or chimeric proteins composed of Rrp6, GFP, and full-length or truncated Mmi1 from plasmids. Transcripts were quantified by RT-qPCR and normalized to *act1*. Error bars represent standard error of three independent samples. **$P < 0.01$; ***$P < 0.001$ compared with cells carrying empty vector (Student's *t*-test). (**C**) Growth profiles of *mmi1-ts3* cells expressing Red1, Rrp6-GFP, Mmi1, or chimeric proteins composed of Rrp6, GFP, and full-length or truncated Mmi1 from plasmids. Ten-fold serial dilutions of cells were spotted on MM medium and incubated at the indicated temperatures. (**D**) Expression of *mei4* mRNA and *ssm4* mRNA in *mmi1-ts3* cells expressing Red1, Rrp6-GFP, Mmi1, or chimeric proteins composed of Rrp6, GFP, and full-length or truncated Mmi1 from plasmids. Cells were grown in liquid MM medium at 25°C and shifted to 37°C for 4 hours. Transcripts were quantified by RT-qPCR and normalized to *act1*. Error bars represent standard error of three independent samples. **$P < 0.01$; ***$P < 0.001$ compared with cells carrying empty vector at 37°C (Student's *t*-test).
(PDF)

**S6 Fig. Interaction between Rrp6 and Mmi1 is essential for meiotic transcript elimination.** (**A**) Expression of *rec8* mRNA, *spo5* mRNA, and *rpl402* PROMPT in *red1Δ* cells expressing Red1, Rrp6-GFP, Mmi1 or chimeric proteins composed of Rrp6, GFP, and full-length or truncated Mmi1 from plasmids in liquid MM medium. Transcripts of each gene were analyzed by RT-qPCR and normalized to *act1*. Error bars represent standard error of three independent samples. *$P < 0.05$; **$P < 0.01$ compared with cells carrying empty vector (Student's *t*-test). (**B**) Growth profiles of *red1Δ* cells expressing Red1, Rrp6-GFP, Mmi1, or chimeric proteins composed of Rrp6, GFP, and full-length or truncated Mmi1 from plasmids. Ten-fold serial dilutions of cells were spotted on MM medium and incubated at the indicated temperatures.

(**C**) Expression levels of chimeric Rrp6-YFP-Mmi1 proteins. Cell extracts were prepared from exponentially growing cells expressing Rrp6-YFP or chimeric proteins composed of Rrp6, YFP, and full-length or truncated Mmi1 from plasmids in liquid MM medium and immuno-blotted with anti-GFP antibody. γ-tubulin was used as a loading control. (**D**) Expression of *rec8* mRNA, *spo5* mRNA and *rpl402* PROMPT in *red1Δ mmi1Δ mei4Δ* cells expressing Red1, Rrp6-GFP, Mmi1, or chimeric proteins composed of Rrp6, GFP, and full-length or truncated Mmi1 from plasmids in liquid MM medium. Transcripts were analyzed by RT-qPCR and normalized to *act1*. Error bars represent standard error of three independent samples. $^{*}P < 0.05$; $^{**}P < 0.01$; $^{***}P < 0.001$ compared with cells carrying empty vector (Student's *t*-test). (**E**) Co-immunoprecipitation of endogenous Mmi1 and chimeric Rrp6-GFP-Mmi1 proteins. Native cell extracts were prepared from exponentially growing cells expressing chimeric proteins composed of Rrp6, GFP, and full-length or truncated Mmi1 from plasmids in liquid MM medium and subjected to immunoprecipitation with anti-GFP antibody. Precipitates and 10% total cell extracts were then immunoblotted with anti-Mmi1 and anti-GFP antibodies. (PDF)

**S7 Fig. Rrp6 is vital for nuclear foci formation of other exosome components.** (**A**) Localization of Dis3 and Rrp4 in *rrp6Δ* cells. *rrp6Δ* cells expressing Dis3-GFP or Rrp4-GFP from the respective endogenous loci were observed during exponential growth in liquid YE medium. Dotted lines indicate the shape of cells. Boxed regions are magnified in Fig 5A. (**B**) Localization of Red1 and Mmi1 in *rrp6Δ* cells. *rrp6Δ* cells expressing Red1-YFP (green) and CFP-Mmi1 (magenta) were examined. Boxed region is magnified in Fig 5B. (**C**) Localization of Dis3 and Rrp4 in *red1Δ* cells expressing Rrp6-YFP, YFP-Mmi1, or chimeric proteins composed of Rrp6, YFP, and full-length or truncated Mmi1. Dis3-mCherry and Rrp4-mCherry were expressed from the respective endogenous loci (green) and YFP-containing chimeric proteins were expressed from plasmids (magenta) in liquid SD medium. Boxed regions are magnified in Fig 5C. Scale bars: 5 μm. (PDF)

**S8 Fig. Mtl1 mediates interaction between Rrp6 and Mmi1.** Expression of *rec8* mRNA and *spo5* mRNA in *mtl1-cs5* cells expressing Mtl1 and Rrp6-GFP-Mmi1 from plasmids. Cells were grown in liquid MM medium at 30˚C and shifted to 20˚C for 2 hours. Transcripts were quantified by RT-qPCR and normalized to *act1*. Error bars represent standard error of three independent samples. $^{*}P < 0.05$; $^{**}P < 0.01$ compared with cells carrying empty vector at 20˚C (Student's *t*-test). (PDF)

**S1 Table. Strains used in this study.** (PDF)

**S2 Table. Primers used in this study.** (PDF)

**S3 Table. Numerical data for all of graphs in this study.** (XLSX)

# Acknowledgments

We thank H. Kato, T. Urano and J. Nakayama for providing the anti-GFP antibody, S. Grewal and the National BioResource Project Japan for fission yeast strains, A. Nakade for technical support, and S. Iwasaki for helpful support.

## Author Contributions

**Conceptualization:** Yuichi Shichino, Akira Yamashita.

**Data curation:** Yuichi Shichino, Yoko Otsubo, Akira Yamashita.

**Formal analysis:** Yuichi Shichino, Akira Yamashita.

**Funding acquisition:** Yoko Otsubo, Akira Yamashita.

**Investigation:** Yuichi Shichino, Yoko Otsubo, Akira Yamashita.

**Methodology:** Yuichi Shichino, Yoko Otsubo, Akira Yamashita.

**Supervision:** Masayuki Yamamoto, Akira Yamashita.

**Validation:** Yuichi Shichino, Yoko Otsubo.

**Writing – original draft:** Yuichi Shichino, Akira Yamashita.

**Writing – review & editing:** Yuichi Shichino, Yoko Otsubo, Masayuki Yamamoto, Akira Yamashita.

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
