## [Decision Letter · Decision Letter 0]

2 Aug 2019

Dear Akira,

Thank you very much for submitting your Research Article entitled 'Meiotic gene silencing factor Red1 recruits the nuclear exosome to YTH-RNA-binding protein Mmi1' to PLOS Genetics. Your manuscript was fully evaluated at the editorial level and by three independent peer reviewers. The reviewers appreciated the attention to an important problem, but raised some substantial concerns about the current manuscript. Based on the reviews, we will not be able to accept this version of the manuscript, but we would be willing to review again a much-revised version. We cannot, of course, promise publication at that time.

If you decide to revise the manuscript for further consideration at PLOS Genetics, please aim to resubmit within the next 60 days, unless it will take extra time to address the concerns of the reviewers, in which case we would appreciate an expected resubmission date by email to plosgenetics@plos.org.

[LINK]

We are sorry that we cannot be more positive about your manuscript at this stage. Please do not hesitate to contact us if you have any concerns or questions.

Yours sincerely,

Juan Mata, Ph.D.

Guest Editor

PLOS Genetics

Gregory P. Copenhaver

Editor-in-Chief

PLOS Genetics

Reviewer 1 requests RNA-FISH studies to address the potential role of Red1 in nuclear retention (point 2), an essential control for addressing the nature interaction between Rrp6 and Mmi1 (RNase treatment, point 4), and an additional control to ensure that the SID mutant is expressed at levels similar to those of other constructs used (point 5). Point 1 is also mentioned by reviewer 3, who suggests specific experiments (see below). I agree that these are important experiments. Points 3 and 6 can be addressed by discussion and clarification of the data.

Reviewer 2 requests investigating the role of MTREC (point 1, I agree this is an important and straightforward experiment). Points 2 and 5 are important controls that will provide further evidence of the model. Points 3 and 4 can be adressed by discussion.

Reviewer 3: Comments 2 and 3 can be addressed by further discussion, and I would be happy with the reviewer’s suggestion of removing the section on self-interaction of Mmi1 being important for degradation (comment 5). Comment 1 (demonstrate direct interaction between proteins) is important and will require further experiments. Point 4 (related to point 1 of reviewer 1) should be addressed experimentally, either by Northern blot (as suggested by the reviewer) or by other methods that the authors find suitable.

Please also address the minor points of reviewers 1 and 2 (points 1, 2 and 4, as 3 is related to comments from reviewer 1).

Please feel free to contact me in some of the suggested experiments are not feasible.

Reviewer's Responses to Questions

**Comments to the Authors:**

Reviewer #1: In this work by Shichino and colleagues, the role of the Red1 protein is characterized. Red1 is known to be part of the MTREC complex, which promotes the degradation of a variety of transcript in fission yeast, including meiotic mRNAs during the mitotic cell cycle. Although Red1 is known to be required for the silencing of meiotic mRNAs, its exact contribution to selective RNA elimination remains poorly understood. In this study, it is shown that Red1 is required for the accumulation of the nuclear exosome to Mmi1 foci and for the copurification of Mmi1 with Rrp6. Using a chimeric fusion protein in which Rrp6 and Mmi1 are expressed as a single polypeptide, it was found that Red1 was now dispensable for silencing mei4 and ssm4 meiotic transcripts, but not a cryptic unstable transcript expressed from the cit6 gene. Using this Rrp6-YFP-Mmi1 fusion, they also show that the self-interaction domain (SID) of Mmi1 was required for the accumulation of the Rrp6-YFP-Mmi1 fusion into nuclear foci. From these data, the authors conclude that the primary role of Red1 in meiotic RNA elimination is to promote recruitment of the nuclear exosome to Mmi1-targeted transcripts.

The major conclusion presented in this work (i.e. that Red1 is important to recruit the nuclear exosome to Mmi1 degradation foci) is potentially interesting. However, key points need to be addressed before publication of this work, especially related to the importance of the Mmi1 nuclear foci for RNA decay and the evolutionarily conserved role of Red1 in nuclear RNA retention.

MAJOR POINTS:

- The study focuses on the colocalization of exosome, Red1, and Mmi1 into nuclear foci that are thought to be degradation centers. However, results in this study questioned the importance of Mmi1 nuclear foci for RNA elimination. Expression of Red1 deletion mutant 196-245 shows complete loss of Rrp6 accumulation in nuclear foci (6%; Fig. 2C). Yet, the Red1 deletion mutant 196-245 only shows 2.5-fold accumulation of mei4 and ssm5 mRNAs, whereas the complete deletion of Red1 (which also shows 3-6% Rrp6 in foci; Fig. 1) leads to 20- and 12-fold accumulations of mei4 and ssm5 mRNAs, respectively (Fig. S2D). These results indicate that the presence of the nuclear exosome to Mmi1 nuclear foci is not absolutely required for degradation of mei4 and ssm4 mRNAs. This is a key point that needs to be clarified.

- The group of Torben Jensen recently showed that the human Red1 homolog, ZFC3H1, is required for the retention of targeted polyadenylated RNAs into nuclear degradation foci (Cell Reports 2018, 23: 2199). An important question that needs to be addressed before publication of this work is whether Red1 is also required for retention of meiotic transcripts in Mmi1 foci. The authors should perform RNA FISH on either ssm4 or mei4 mRNAs in red1-null cells to determine whether these RNAs are still tethered to Mmi1 foci or are now exported to the cytoplasm. This is important to dissect Red1’s role in nuclear exosome recruitment and nuclear RNA retention.

- There are major inconsistencies in the RT-qPCR analysis of RNA levels presented throughout this study. For instance, whereas the mei4 mRNA is upregulated 20-fold in red1-null cells in Fig. S2D, it only shows 3-fold accumulation in Fig. 3B in the same deletion. The same is true for ssm4: 12-fold accumulation in red1 mutant in Fig. S2D, but 4-fold in Fig. 3B. The cit6 PROMT is up 15-fold in the red1 mutant in Fig. S2E, but shows 5-fold accumulation in the same mutant in Fig. S2F. This needs to be addressed.

- Fig. 3A and S3 report physical association between Rrp6 and Mmi1. Because Mmi1 is an RNA-binding protein, it is very important to address whether the copurification of Rrp6 and Mmi1 is mediated by protein-protein interactions or depends on RNA by performing the coIP with and without benzonase. Alternatively, they could use RNA-binding defective mutants of Mmi1 such as Y352F or Y466F.

- The authors compare the molecular phenotypes of various Rrp6-YFP-Mmi1 fusions: full-length (FL), delta YTH, and delta SID. The delta SID mutant shows defective localization to nuclear foci and is unable to rescue RNA silencing in the red1 mutant. Accordingly, it is very important to show that this mutant (delta SID) is expressed at levels similar to the FL and delta YTH by Western blotting.

- It is not explained why analysis of Mmi1 function using the Rrp6-YFP-Mmi1 fusion was done in the mmi1-ts3 mutant and not in the deletion mutant, as done in Fig. 1 and Fig. S2. The use of the mmi1 deletion mutant represents a cleaner genetic system than the ts mutant.

MINOR POINTS:

- p. 6, lines 5-7: work by the Leatherwood (Chen et al. 2011 PLoS ONE) and Bachand (St-André et al. 2010, JBC) labs should be cited for studies on Pab2 and meiotic RNA elimination.

- p. 5, lines 5-8: the Mmi1 binding motif was also uncovered by a CRAC analysis of Mmi1 from the Vasiljeva lab (Cell Rep, 2015). Should cite.

- In Fig. 2A, the authors could show the various Red1 deletions using schematic. This would ease comprehension by readers.

Reviewer #2: In this study, Shichino and colleagues investigate the relationship between the formation of exosome nuclear foci and the degradation of meiotic transcripts in S. pombe vegetative cells. The authors propose a role for the Red1 subunit of the MTREC complex in physically bridging the YTH family RNA-binding protein Mmi1, which targets meiotic transcripts, to the nuclear exosome, for efficient degradation. Previous studies showed that Red1 associates with and connects Mmi1 to the nuclear exosome (Sugiyama and Sugiyama, 2011; Lee et al., 2013; Egan et al., 2014; Zhou et al., 2015).

Specifically, the authors show in a series of microscopy and genetic experiments that i) an N-terminal region of Red1 (residues 196 to 245) is required for its own accumulation and the enrichment of exosome components in Mmi1-containing nuclear foci, ii) expression of a chimeric construct linking Mmi1 to the Rrp6 subunit of the exosome bypasses the requirement of Red1 for exosome foci formation as well as meiotic mRNA degradation.

The experiments presented are mostly convincing and well performed. However, it remains questionable whether the role of Red1 in bridging Mmi1 to the exosome is specific or underlies a function of MTREC as a whole. Biochemical experiments are also needed to support the authors’ conclusions and apparent discrepancies should be discussed as well. Below, issues are outlined as major and minor points.

Major points:

1/ An important aspect of the proposed model is that Red1 physically links the exosome to Mmi1 to promote efficient meiotic mRNAs degradation. Is this function specific to Red1 or dependent on MTREC? It would be informative to analyze a mutant of the core MTREC component Mtl1 in exosome foci formation and in meiotic mRNAs degradation in the presence of the chimeric Mmi1-Rrp6 construct.

2/ The red1(∆196-245) mutant is defective for the formation of exosome and its own foci (Figure 2). This might suggest that the interaction between Mmi1 and the exosome is lost in this context. Another possibility is that Red1(∆196-245) is defective for its association with Mmi1, Rrp6 and/or Mtl1. Biochemical evidence should be provided.

3/ The red1(∆196-245) mutant only marginally impacts meiotic mRNA degradation (Figure S2), suggesting that exosome foci formation may not be a critical determinant for transcripts degradation. The authors should discuss these apparent discrepancies in their model.

4/ Data presented in Figure 1A/S1A indicate that Mmi1 is only partially involved in Rrp6 foci formation, contrary to Red1. This indicates that the exosome forms foci independently of Mmi1 and further questions the correlation between exosome foci formation and meiotic mRNA degradation.

5/ The authors show that Rrp6-GFP-Mmi1-YTH∆ suppresses meiotic mRNA accumulation in red1∆ but not red1∆ mei4∆ mmi1∆ cells, suggesting that Rrp6-GFP-Mmi1-YTH∆ requires endogenous Mmi1. The authors should verify that endogenous Mmi1 associate with their chimeric construct in a SID-dependent manner. Also, one prediction is that the Mmi1 cofactor Erh1 should be required for the chimeric construct to function.

Minor points:

1/ A quantification of the number of exosome nuclear foci in wt and mutant cells would be informative, especially since mei4∆ mmi1∆ cells appear to have less dots.

2/ From the data presented (1B, S1B), there are very few Dis3 and Rrp4 dots outside the nucleolus. The percentage of foci frequency is not supported by the data. Could it be that Rrp6 is tethered to foci independently of the core exosome?

3/ Data in Figure 2B indicate that Red1 signals for the ∆2-347, ∆196-347 and ∆196-245 mutants are less intense. The authors should verify Red1 expression levels to exclude the possibility that that the lower frequency of dots is due to lower protein amount in the mutants.

4/ Expression analyses of additional CUTs should be provided to extend the results described for the cti6 PROMPT.

Reviewer #3: In this manuscript, Shichino et al., investigate role of Zn finger protein Red1, one of the factor connected to nuclear exosome in fission yeast by the previous studies, in elimination of meiotic mRNAs. They propose that Red1 plays a role of connecting RNA binding sequence specific protein Mmi1 with the exosome subunit Rrp6 and test this hypothesis using several complementary approaches.

Although the study aims to address an important question in the field of RNA degradation related to the specificity of targeting, I feel that further evidence is needed to support the claims before I could recommend this manuscript for publication.

Comments:

1. Throughout the manuscript, authors refer to the previously published work to support that red1 interact with rrp6 and mmi1 directly (for example page 12, 1st paragraph). The Co-IP of native complexes can not be used as an evidence for direct interaction. The current study does not provide this evince either. The proper biochemical experiments demonstrating direct interaction for recombinant proteins would benefit the manuscript.

2. The importance of the nuclear foci for degradation of meiotic transcripts is not clear to me from the data presented. Data presented in Figure S2D does not show accumulation of meiotic transcripts in red1 delta 196-245 where foci are not formed.

3. In Figure S4E expression of rrp6 and mmi1 separately rescues red1 delta growth defect equally well to the expression of the Rrp6-Mmi1 fusion protein suggesting that while having more Rrp6 and Mmi1 can rescue the phenotype physical link between these may not be important.

4. Data in Fig 3B are quite central to the manuscript as they are meant to demonstrate that physical link between Rrp6 and Mmi1 is important for elimination of meiotic transcripts. However, the differences shown in this panel are not very striking, the numbers greatly varied between experiments (barely 3 fold difference for mei4 RNA in red1 delta in fig 3B compared to more than 20 fold difference in Figure S2D) and analyses is limited to only a couple of meiotic transcripts. I suggest extending analyses to more meiotic RNAs and presenting Northern blots and their quantifications.

5. Section on self interaction of Mmi1 being important for degradation is the weakest part of the manuscript. I am sure that the manuscript benefit from this section at all. All the conclusions drawn based on indirect evidence and not convincing.

**Have all data underlying the figures and results presented in the manuscript been provided?**

Reviewer #1: Yes

Reviewer #2: Yes

Reviewer #3: Yes

PLOS authors have the option to publish the peer review history of their article (what does this mean?). If published, this will include your full peer review and any attached files.

Reviewer #1: No

Reviewer #2: No

Reviewer #3: No

---

## [Decision Letter · Decision Letter 1]

23 Dec 2019

Dear Akira,

Thank you very much for submitting your Research Article entitled 'Meiotic gene silencing factor Red1 recruits the nuclear exosome to YTH-RNA-binding protein Mmi1' to PLOS Genetics. Your manuscript was fully evaluated at the editorial level and by independent peer reviewers. The reviewers appreciated the attention to an important topic but identified some aspects of the manuscript that should be improved.

We therefore ask you to modify the manuscript according to the review recommendations before we can consider your manuscript for acceptance. Your revisions should address the specific points raised by reviewer 2 (no additional experients required).

[LINK]

Yours sincerely,

Juan Mata, Ph.D.

Guest Editor

PLOS Genetics

Gregory P. Copenhaver

Editor-in-Chief

PLOS Genetics

Please address the minor comments raised by reviewer number 2 (no extra experiments required).

Reviewer's Responses to Questions

**Comments to the Authors:**

Reviewer #1: The authors have satisfactorily addressed my concerns. The manuscript has now improved and is now suitable for publication.

Reviewer #2: In the revised version, the authors provide additional data that globally improve the manuscript. Below are minor comments:

1) From data shown in Figure 6A and S8A, no Rrp6 foci are visible in the mtl1-cs5 mutant while the authors claim that there is only a mild effect on their formation. Images where such dots are present should be shown, consistent with the quantification.

2) The authors now show that Mtl1 is required for exosome foci formation and that the chimeric Mm1-GFP-Rrrp6 protein suppresses ectopic expression of meiotic mRNAs in the mtl1-cs5 mutant. Since there is no in vitro evidence that recombinant Mtl1 associates or not with Mmi1 and Rrp6, the authors should tune down the emphasis on Red1 in the title and the abstract, as they did in the discussion (“Our observations also imply that the MTREC/NURS complex, rather than Red1 alone, may link Mmi1 to the exosome”).

Reviewer #3: The authors have addressed most of the issues raised by the reviewers

**Have all data underlying the figures and results presented in the manuscript been provided?**

Reviewer #1: Yes

Reviewer #2: Yes

Reviewer #3: Yes

PLOS authors have the option to publish the peer review history of their article (what does this mean?). If published, this will include your full peer review and any attached files.

Reviewer #1: No

Reviewer #2: No

Reviewer #3: No

---

## [Editor Report · Decision Letter 2]

3 Jan 2020

Dear Akira,

We are pleased to inform you that your manuscript entitled "Meiotic gene silencing complex MTREC/NURS recruits the nuclear exosome to YTH-RNA-binding protein Mmi1" has been editorially accepted for publication in PLOS Genetics. Congratulations!

Yours sincerely,

Juan Mata, Ph.D.

Guest Editor

PLOS Genetics

Gregory P. Copenhaver

Editor-in-Chief

PLOS Genetics

Comments from the reviewers (if applicable):

**Data Deposition**

http://datadryad.org/submit?journalID=pgenetics&manu=PGENETICS-D-19-01079R2

**Press Queries**

---

## [Editor Report · Acceptance letter]

27 Jan 2020

PGENETICS-D-19-01079R2 

Meiotic gene silencing complex MTREC/NURS recruits the nuclear exosome to YTH-RNA-binding protein Mmi1 

Dear Dr Yamashita, 

We are pleased to inform you that your manuscript entitled "Meiotic gene silencing complex MTREC/NURS recruits the nuclear exosome to YTH-RNA-binding protein Mmi1" has been formally accepted for publication in PLOS Genetics! Your manuscript is now with our production department and you will be notified of the publication date in due course.

With kind regards,

Matt Lyles

PLOS Genetics

On behalf of:
